# FossilSketch: A novel interactive web interface for teaching university-level micropaleontology

Category: Research

## ABSTRACT

Micropaleontology studies fossils that are very small and require the use of a microscope. Micropaleontologists use microfossils to analyze data critical for estimating future sea level rise, understanding the causes of past climate upheavals, and finding economically important resources like oil and gas. This subject is taught as part of some geology classes at the undergraduate and graduate university level, but training in this field is time-consuming and less classroom time is typically devoted to the topic. Although demand for geoscientists is projected to grow, fewer students are exposed and trained in micropaleontology. Geosciences currently need micropaleontologists as the population of experts is declining. While interactive math and engineering web interfaces are recently becoming more common, a similar system that provides students with a repository of knowledge and interactive exercises in the micropaleontology space was lacking. To address this problem of training students in micropaleontology, we developed FossilSketch: a web-based interactive learning tool that teaches, trains, and assesses students in the basics of micropaleontology. The interface we developed contains various interactions and a new template-based system to check drawn shape accuracy helps students learn characteristic features of microfossils. Our evaluation included deploying this system to 32 students in an undergraduate geology class at our university. The accompanying user study results indicate that FossilSketch is an engaging educational tool that can be deployed alongside the classroom for in-class and at-home learning. Student feedback together with our recorded submission data for various exercises suggests that FossilSketch is an effective online learning tool that serves as a helpful reference for class activities, allows for remote learning, presents helpful and engaging interactive games, and encourages repeat submissions.

**Index Terms:** Human-centered computing—Interactive systems and tools; Information systems—Web applications; Applied computing—Interactive learning environments; Applied computing—Earth and atmospheric sciences;

## 1 INTRODUCTION

The fossil remains of micro-organisms preserved in modern and ancient sediments play key roles in determining the ages of geologic records, reconstructing ancient environments, and monitoring modern ecosystem health. However, training undergraduates to identify these microfossils is time-intensive and most students are not exposed to this tool in their courses. Core geoscience courses that reach all majors rarely include micropaleontology, the study of microfossils, because contact hours are not sufficient to train students at the necessary level of detail. Student training in micropaleontology has declined over the last several decades as the field of geology has broadened and micropaleontology has been replaced by other methods [6, 48]. Thus, although the geosciences currently need micropaleontologists because the population of experts is aging, few students are trained to use this tool [40].

To enable and enhance training of undergraduates in the basics of micropaleontology in remote, hybrid and in-class conditions, we developed FossilSketch, an interactive digital tool that introduces students to micropaleontology through educational videos, sketch-based exercises and mini-games focused on microfossils and their applications in geosciences. FossilSketch, depicted in Figure 1, makes use of a modified version of the existing Hausdorff template

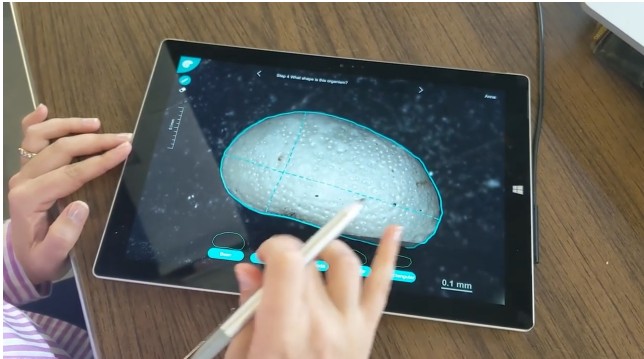

Figure 1: A participant using the FossilSketch educational web app.

matching technique to support automated grading of activities involving sketching microfossil outlines. This lightweight recognition technique is able to calculate cumulative distance between resampled points from the input sketch and only a single instructor-provided template. This recognition system effectively acts as a shape accuracy algorithm, returning the cumulative distance as an index of dissimilarity between a student-provided sketch and the instructor-provided template. This forms the basis of the system's recognition technique that is used in the identification exercises designed for two microfossil groups, Foraminifera and Ostracoda. The paper also outlines the other interactive games and assessments that underlie the FossilSketch system.

## 2 BACKGROUND INFORMATION

### 2.1 Micropaleontology

Micropaleontology is a critical tool for determining the ages of sedimentary rocks for both industrial (e.g., oil exploration) and scientific applications [28]. Microfossil species are also sensitive to specific environmental parameters and are often used to reconstruct past changes in ocean temperature, coastal sea-level, and seafloor oxygenation [36]. Further, microfossils are used in modern, real-time, environmental monitoring because they respond quickly to environmental change [12]. Despite their increasing usefulness, training students in micropaleontology has declined.

Foraminifera and Ostracoda are two of the most commonly used microfossils in industrial, environmental, and scientific applications; these are also some of the larger microfossils, which allows students to view them with standard stereoscopes. Foraminifera are amoeboid protists with shells made of calcium carbonate or agglutinated sediment grains and are often abundant in marine environments [6]. Ostracoda are micro-crustaceans with a bivalved calcareous carapace that are found in all aquatic environments from fresh water lakes to to the deep-sea [6]. The morphology of species in both groups is closely related to the environments in which they live [22, 29, 43] and these two groups are often used in species-specific geochemical studies [25], thus accurate identification is important for using this tool. FossilSketch application focuses on these two groups of microfossils.

Accurate identification of species is the crucial first step in all

applications of microfossils. Sketching is critical in understanding the morphological differences because it helps students internalize the characteristic features and better understand them by connecting their sketch to the specimen. Researchers find that sketching benefits learning in a wide range of disciplines, from human anatomy and biology to engineering, geography and math [9, 20, 21, 39, 45]. However, one of the challenges in teaching micropaleontology is the amount of individual feedback students need on their sketches to ensure they are learning the correct features for identification.

## 2.2 Related Works

### 2.2.1 Geoscience Educational Tools

The geosciences have rapidly adopted online and remote-based educational tools over the last five years. The popularity of online learning platforms has led to the development of online resources, new pedagogical practices, and course curricula (e.g., [5, 10, 11]). Successful examples that integrate technology into geoscience classes include high resolution digital imaging for mapping and documenting geological outcrops, 3D virtual simulations, and digitalization of fossil collections [8, 15, 27]. For laboratory-based courses, scholarship has primarily focused on accessibility for students with visual disabilities at the introductory level [13] whereas field-based course literature on accessibility has mostly focused on inclusive practices to better serve students with mobility disabilities [13].

Some of the successful software used in geoscience education include the following. Researchers at Northwestern University and IBM pioneered sketching software uses in geoscience with the CogSketch application and a series of 26 introductory geoscience worksheets about key geoscience concepts [20]. CogSketch aids students in solving discipline-specific spatial problems while providing instructors with insights into student thinking and learning. Real-time feedback identifies erroneous sketch features, and helps students reconsider and correct them. Milliken developed tutorials to study sandstone petrology at the University of Texas at Austin using a "virtual microscope" [33]. Students are able to practice identification of a wide array of sandstone components outside of the laboratory and independent of the instructor. They found student attainment of petrography skills improved with tutorial use.

As for micropaleontology, researchers note a lack of human experts and decline in micropalentology training [14, 26, 34], however, most software development has been aimed at automated identification of microfossils. The earliest attempts lacked accuracy and were not fully automated [7, 46]. More recent approaches to automated micropaleontology identification software usually focuses on machine learning and uses 3D models for planktic and benthic foraminifera identification [14, 26, 34]. Their results indicate that current image classification techniques perform identifications comparably to human experts [34].

Several large microfossil databases were built that include taxonomic hierarchy data, images, ecological characteristics and geographical distribution, as well as type species information (e.g., for Ostracoda: Modern Podocopid Database [17]; World Ostracoda Database (WorMS) [1]; for Foraminifera: World Foraminifera Database, (WorMS) [2]; Foraminifera Gallery, (Foraminifera.eu [44]). However, these online resources are designed for an advanced user and are difficult to use for entry level specialists and students without instruction on microfossil morphology.

To summarize, there is clearly a need and growing interest in developing automated AI methods for microfossil identification due to decline in human experts numbers. We believe that developing educational software on Foraminifera and Ostracoda would be a more efficient approach to solving the problem of the lack of human experts. Thus, designing novel, universally accessible, and academically rigorous educational tools is a highly relevant task for undergraduate geoscience education.

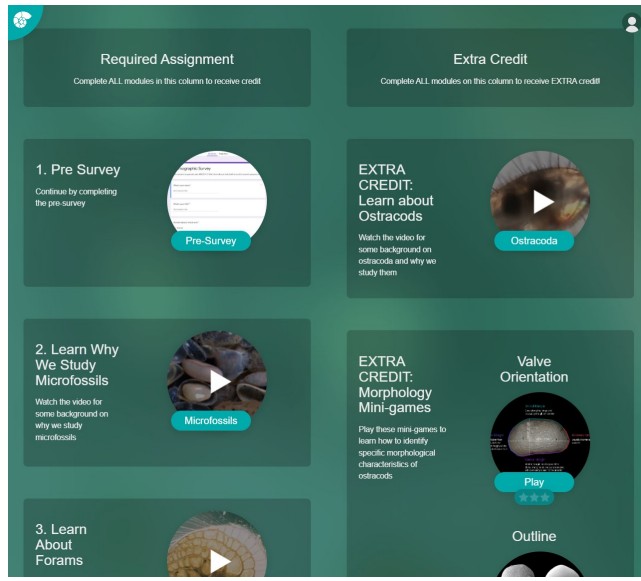

Figure 2: After students log in, they are shown the landing page. Modules are divided into two columns, with required sections on the left and optional, extra credit, on the right.

### 2.2.2 Digital Sketch Recognition in the Classroom

Sketching activities in the classroom have pedagogically been linked to enhanced student creativity and learning [35, 37, 41, 52, 53]. Studies have confirmed that information retention and learning outcomes are significantly improved when engaging in drawing and writing activities vs. using a keyboard as the primary input modality [35]. Sketch-based learning tools have been linked to a higher retention of information and improved skill compared against students who do not learn with sketch-based activities [23, 54]

Early gesture recognition systems developed by Rubine [47] have led to improved recognition systems including template-matching algorithms from the "Dollar" family of recognizers [3,4,50,51,55] that produced lightweight recognition systems easily added to existing software. The "Dollar" recognizers perform classification tasks by using different methods of calculating distance from user-generated input compared against several samples of trained data. Despite these recognizers being used for classification techniques rather than grading sketch accuracy, we use this work as a basis for our recognition system. Both feature-based classification techniques and template matching techniques were later expanded into more robust systems for scaffolded recognition via systems like *PaleoSketch* [42] and *LADDER* [24], the second of which is notable for its integration of domain-specific shapes to better describe relationships between sketch properties to assist in recognition. More recent works like *nuSketch* [19] and *COGSketch* [18] integrate sketch recognition algorithms into educational tools to assist with the learning experience to measurable success.

*Mechanix* [38, 49], *Newton's Pen* [32] and *Newton's Pen II* [31], and *Physics Book* [16] are systems specifically written to leverage the educational advantage of drawing and sketching into the core interactions of their tools. Indeed, these systems serve as the primary conceptual basis from which FossilSketch is designed. We aimed at adapting the educational techniques presented by these tools into the domain of micropaleontology in the classroom. This led to a variety of changes and design considerations taken in the teaching approach outlined in the next section.

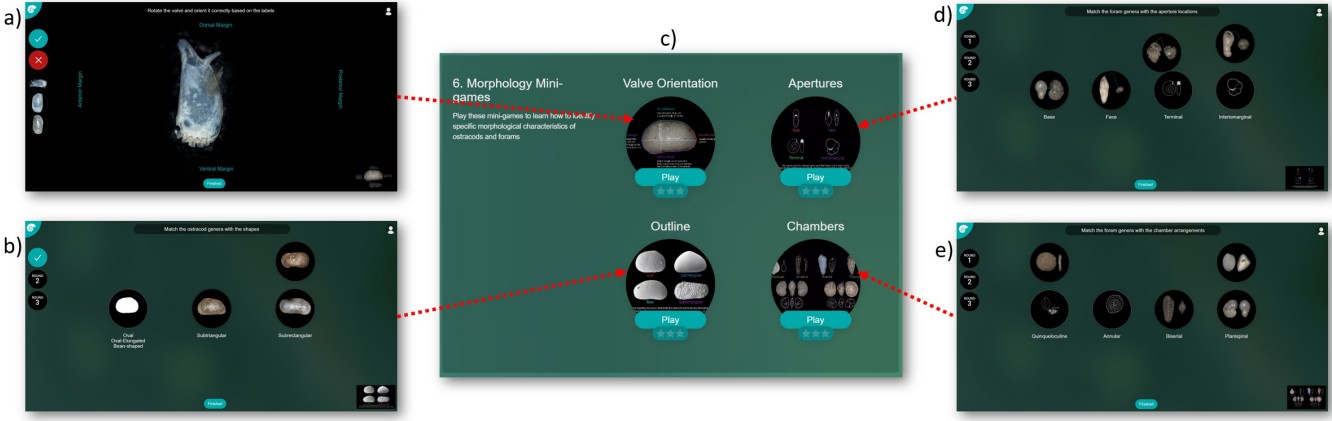

Figure 3: Overview of the activities in FossilSketch. a) is an example of an Ostracoda Orientation Game; b), d) and e) are examples of Foraminifera Matching Games; c) is a cropped screenshot of one of the modules from the FossilSketch landing page. Red arrows indicate which sub-figure belongs to which game, but arrows are not part of the FossilSketch interface and for illustrative purposes only.

## 3 INTERFACE DESIGN

### 3.1 Design Considerations

FossilSketch is a web-based educational tool for teaching students techniques for identifying microfossils. Educational materials for FossilSketch were developed to supplement various geoscience courses in the College of Geosciences at [author institution redacted for review]. Traditionally, undergraduate students learn about micropaleontology through lectures, diagrams, specimens viewed through a stereoscope, and hand-sized models in upper-level courses for geology majors. To allow for comparison between traditional and FossilSketch-based classes, we developed analogous educational materials for both groups. FossilSketch educational materials include the following: 1) educational videos; 2) instructional mini-games; 3) practice exercises; and 4) assessments. All four types of activities consist of content specifically created for FossilSketch and tailored to support the educational exercises in traditional and FossilSketch-based courses.

Exercises were developed based on the course learning objectives, the microfossil collections available, and the expertise of [co-author names redacted for review]. The level of difficulty and number of activities varied depending on whether the course is lower or upper division and whether the course primarily serves geoscience or non-geoscience majors. The landing page for each course also varied depending on the teaching goals and the activities assigned to students.

In Fall 2021, FossilSketch was deployed in Geol 208 ("Life on a Dynamic Planet"), a lower division undergraduate course where most students are not geology majors. Students were given access to FossilSketch 5 days before the in-person laboratory session during which one hour of laboratory time was devoted to FossilSketch activities. Students were required to complete activities for Foraminifera and could complete the Ostracoda activities included in a separate column of modules for extra credit.

### 3.2 Interface Description

#### 3.2.1 Landing Page

The FossilSketch website initially prompts new and returning users to log in with their credentials. To ensure data integrity, new user registration is limited via a registration code assigned to each group of students who are part of the study, with each group being assigned a different code. Test accounts and external evaluators were assigned special login credentials and their activities were not recorded as part of the data collection.

After the participants log in, modules are listed in the order in which they are meant to be completed. Modules were added, modified, or removed depending on the class or activity in which FossilSketch was deployed. The landing page used in our current study is shown in Figure 2.

The self-contained nature of the exercises and the flexibility of the landing page interface offers the versatility of adding new exercises and rearranging the website experience depending on the course learning objectives.

#### 3.2.2 Educational Videos

Educational videos were created specifically for FossilSketch and were written to provide introductory information to help contextualize concepts covered in the rest of the FossilSketch's activity types. When users click on these modules, an overlay with an embedded YouTube link is displayed. Students are free to change playback with the standard embedded YouTube video controls and the overlay can be dismissed at any time by clicking outside of the video area. No progress data is recorded for this type of activity.

FossilSketch is intended to augment instructor lectures, meaning the videos are not intended to serve as a replacement for lecture material as is usually the case with typical instructional videos in an online learning interface. The FossilSketch system uses instructional videos to provide necessary information for students to engage with the rest of the modules if the students have not yet received instructor lectures, while at the same time emphasising concepts most directly relevant to the activities if they have attended in-depth lectures in the classroom.

#### 3.2.3 Instructional Mini-Games

FossilSketch integrates various kinds of interactive instructional tools. In order to improve student comprehension of microfossil identification, we broke identification tasks into small, minigame, tasks. Students were able to repeat tasks for mastery. Each minigame consists of one or more types of interactions intended to highlight the visual-morphology aspect of learning about microfossil identification.

**Matching Games** require the participants to match morphological features, such as outline shape for Ostracoda, or morphotype and type of chamber arrangement for Foraminifera. At the beginning of the game the students are presented a reference image that lists each morphotype along with a sketched example, and students are able to return to this reference image again, when needed, by clicking on the zoomed-out image on the bottom right corner of the

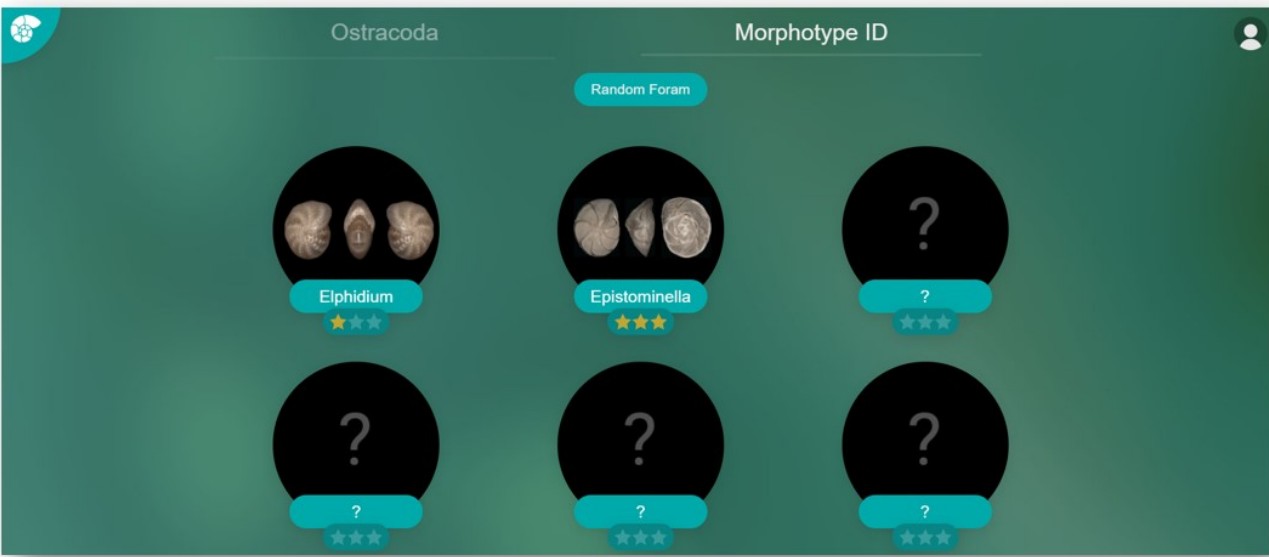

Figure 4: Menu of the morphotype ID exercises. Students pick from any of the unidentified morphotypes marked with a "?", and afterwards are shown their performance on a 3-star rating system.

screen. When the game starts, the screen displays a small number of draggable "discs" or rectangular "cards" with actual microfossil photomicrographs that the user can move into slots with sketched categories for each feature used in this game. At the moment, four different mini games are created with this kind of interaction: Ostracoda lateral outline identification; Foraminifera apertures, chamber arrangement, and Foraminifera morphotypes identification.

All matching games include three rounds, with each level contributing to a final star score. The Foraminifera apertures and chamber arrangement mini games randomly pull images of Foraminifera from the database for matching to the corresponding apertures and chamber arrangement types, with each round of game having four cards to match. In the morphotype mini game, the number of draggable items and slots in later rounds increases from 4 in the first round, to 8 in the third round to increase difficulty. Students receive star rating form zero to three on how many rounds they got correctly in the first attempt.

**Orientation Games** integrate a rotation interaction to help students gain an understanding of how to correctly orient the ostracod valve for identification. An ostracod valve has four sides: dorsal, ventral, posterior, and anterior margins/side. This game starts with a general description of each of these margins to help students gain an intuition of how to identify each side of an ostracod. The user is tasked with rotating an ostracod to its position with the dorsal side up and all of its sides correctly labeled. To simplify the interaction, students rotate in one direction 90 degrees at a time by clicking or tapping once on the ostracod that is displayed on the center of the screen. When the student believes that the ostracod is oriented correctly, they submit their answer by selecting the "Finished" button on the center bottom of the screen.

Like with matching games, orientation games are divided into three rounds. In this case, each round consists of one ostracod valve that needs be rotated into correct orientation. Answers are marked "correct" if they are rotated correctly the first time the "Finished" button is clicked. Like in the matching games, students will need to correct their answer if it is incorrect to move onto to the next round, but the answer will still be marked incorrect. Students are encouraged to use the knowledge gained by correcting their wrong answer to try the exercise again to receive full credit for their answers and receive a 3 star rating.

### 3.2.4 Identification Exercises

In micropaleontology, microfossils are picked from sediment samples and the obtained variety of different species represents an assemblage characteristic of the sample and may indicate the environmental setting or geologic age of the sample. A micropaleontologist would identify the species of microfossils in this assemblage based on their morphology, or their characteristic features. One of the goals of this interface is to demonstrate to students the various applications of microfossils in geosciences. Primarily, FossilSketch offers a scaffolded learning experience to guide students through the steps needed to identify microfossils and their morphological characteristics.

For the undergraduate course Geol 208, students identified foraminiferal morphotypes, and Ostracoda genera (as an extra credit). Students are first presented with a menu depicted in Figure 4. Once chosen, the Foraminifera morphotypes identification steps can be seen in Figure 5 and are the following: 1) sketch the outline of the foraminifer image on the left; 2) sketch the outline of the foraminifer image in the center; 3) choose the overall shape of the organism from a menu; 4) choose the type of chamber arrangement from the menu; 5) find and click on the aperture location in the center image; 6) identify a morphotype based on the selected features. The Ostracoda genera identification exercise steps are shown in Figure 6 and include: 1) sketch the maximum length of the valve; 2) sketch the maximum height of the valve; 3) identify right vs left valve; 4) sketch the outline of the ostracod valve; 5) choose the type of outline from the menu; 6) measure approximate size of the valve and choose the size range from the menu; 7) choose the types of ornamentation; 8) identify an ostracod genus based on the selected features.

Within each exercise the types of interactions are described below:

**Sketching interactions** (steps 1-2 for Foraminifera, and steps 1-2 and 4 for Ostracoda) help students retain and understand the various shapes and outlines they observe in different microfossils.

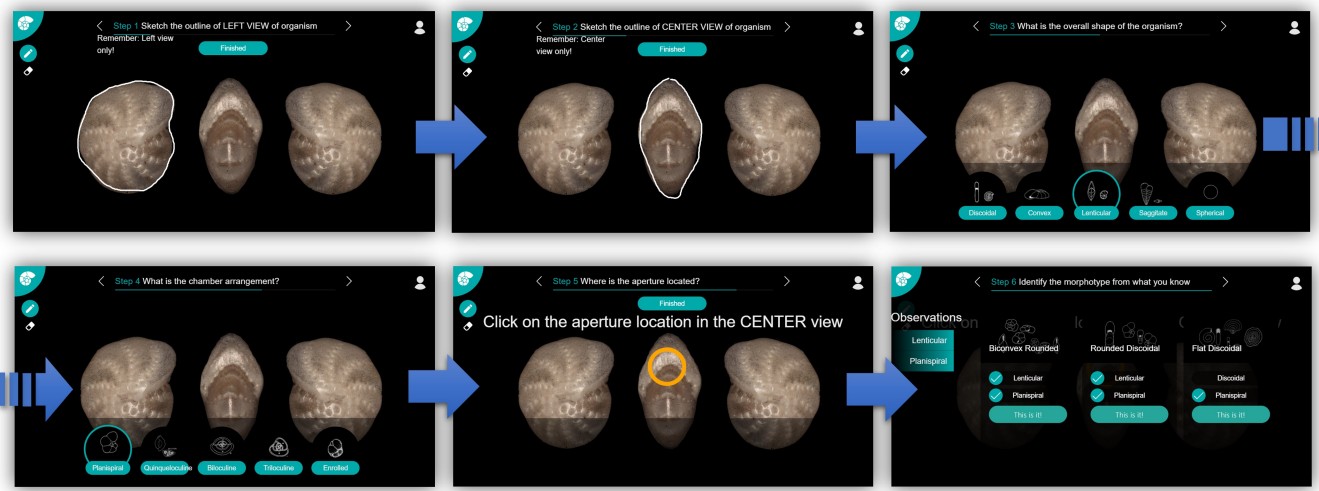

Figure 5: Step by step morphotype ID exercise, starting at the top-left screen and ending at the bottom-right, it includes the following steps: 1) sketch the left view of the organism, 2) sketch the middle view, 3) pick the overall shape, 4) pick the chamber arrangement, 5) click on the area of the aperture location, and 6) draw your conclusion - identify Foraminifera morphotype.

It is the primary method of interaction after which the project is named. Sketching interactions integrate functionality from a library called paper.js to deliver flexible drawing interactions. Although the system is intended to be used with styli and touch to most naturally resemble a sketching activity, it is also possible to draw with a mouse or trackpad. Drawing interactions are usually integrated as the first steps of both kinds of identification exercises, as the overall shape of the sample is critical in identifying the microfossil.

The FossilSketch system checks for correctness using a template-matching recognition heuristic. The template recognizer coded specifically for FossilSketch uses the Hausdorff-distance template matching technique as a baseline, implemented to act as a shape accuracy algorithm. We first resample both, the template and the input sketch, to a lower sampling rate with roughly equidistant points. The formula followed for calculating the interspace distance is:

$$S = \frac{\sqrt{(x_m - x_n)^2 + (y_m - y_n)^2}}{c = 256} \tag{1}$$

where $c = 256$ is a constant empirically derived to adjust the distance between the points for optimal calculation of the distance metric. With the distance calculated, the sketch is resampled using the technique outlined in Algorithm 1.

---

**Algorithm 1** Resampling Technique

---

**Require:** Point list *path*, distance $S$
**Ensure:** Re-sampled point list *out*
    $D \leftarrow 0$
    **for** $i$ in *path* **do**
        $BetweenDist \leftarrow \sqrt{(x_{i+1} - x_i)^2 + (y_{i+1} - y_i)^2}$
        $D \leftarrow D + BetweenDist$
        **if** $D > S$ **then**
            $D \leftarrow BetweenDist$
            $out \leftarrow$ new point $(x_i, y_i)$
        **end if**
    **end for**

---

This iterates through each point in the provided path and gradually adds the distance between the current point and the next until the predetermined distance $S$ is reached, which is where the point will

be placed. The algorithm repeats this process for every point in the input path.

We then iterate through each point in the input sketch, compare it with the corresponding point for the template sketch, and calculate the Euclidean distance between the two. Total distance is calculated across all the compared points and the cumulative sum is the overall "distance" between a template and the student input (see Figure 7). If the average deviation of the points is greater than the pixel with of the canvas divided by a constant, we would determine that the input sketch is too different from the template sketch. This constant was empirically determined after internal testing to match the desired student experience; students are meant to provide a relatively accurate, but not perfect, recreation of the template. This algorithm is outlined in Algorithm 2.

---

**Algorithm 2** Compare Sketches

---

**Require:** Student $Spath$, template $Tpath$
**Ensure:** Boolean *result*
    $totalDeviation \leftarrow 0$
    **for** $i$ in $Spath$ **do**
        $closestDistance \leftarrow INF$
        $longestIndex \leftarrow 0$
        **for** $j$ in $Tpath$ **do**
            $tempDist \leftarrow$ distance between $Spath_i$ and $Tpath_j$
            **if** $tempDist < closestDistance$ **then**
                $closestDist \leftarrow tempDist$
                $closestIndex \leftarrow j$
            **end if**
        **end for**
    **end for**
    $avgDeviation \leftarrow \frac{totalDeviation}{spathlength}$
    $cwidth \leftarrow$ pixel width of canvas
    **if** $avgDeviation > \frac{cwidth}{70}$ **then**
        $result \leftarrow$ True
    **else**
        $result \leftarrow$ False
    **end if**

---

The template sketches are provided by [co-author names redacted

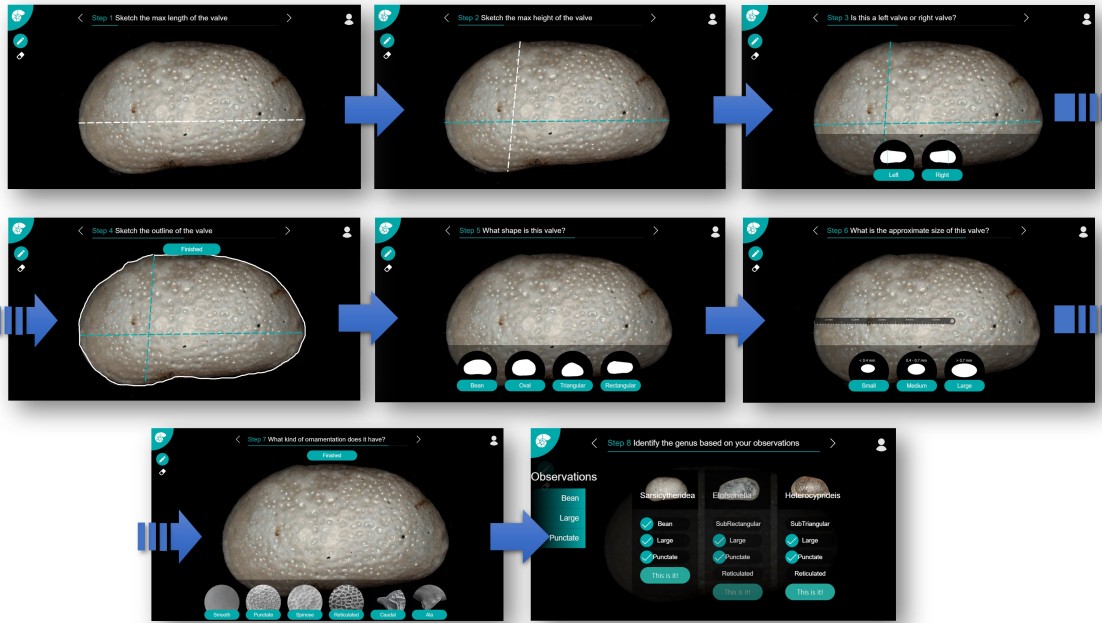

Figure 6: Step by step of the ostracod ID exercise, starting at the top-left screen and ending at the bottom-right, it includes the following steps: 1) draw the max length of the ostracod, 2) draw the max height, 3) identify if it is a left or right valve, 4) sketch the outline of the ostracod, 5) choose the overall shape, 6) determine the length, 7) choose if the valve has ornamentation, and what are the ornamentation features, 8) draw your conclusion - identify Ostracoda genus.

for review] and coded directly into each foraminifer or ostracod image. Every foraminifer in FossilSketch has a database containing template sketch data the outline for its left view, its center view, its largest chamber, and coordinates for the location of the opening - aperture. The last item is used in the interaction labeled "Pointing Interactions" in this section. For every ostracod in a database there is a template sketch data for the outline, maximum length, and maximum height.

**Identification interactions** (steps 3-5 for Foraminifera, and steps 3, 5-6 for Ostracoda) are presented to students as a horizontal multiple-choice menu along the bottom of the screen, and the student is asked to identify one of several characteristic features of the microfossils. For instance, the student might be asked *"what is the overall shape of the organism?"* and the possible answers might be *"vase-like", "convex", "low-conical", "spherical"* and *"arch"* among others. With each option, a sample sketched outline of each shape is shown, but it is important to note these are sketched examples and not photorealistic depictions of the choices. The student is tasked with remembering the particular physical properties of each characteristic feature rather than simply matching the pictures with the closest choice. Of these, one is the correct answer. In this part of the exercise, the student does not receive immediate feedback as to the correctness of this particular question, since all of these answers are summarized for the student to use to identify the foraminifer's morphotype or ostracod's genus.

**Pointing interactions** (step 5 for Foraminifera) are simplified forms of "sketching interactions" that require students to click once in a general area of interest, and FossilSketch checks if the identified location is correct. Specifically, this interaction is used to identify the general location of the aperture of a given foraminifer. The student is asked to click once in the region where they believe the aperture is. Each foraminifer in the FossilSketch database contains data on a rectangular region that points to the general area of its

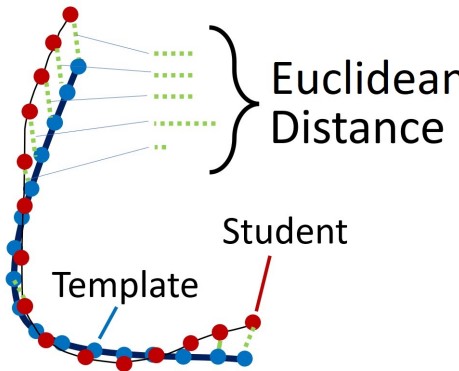

Figure 7: To grade answers, FossilSketch resamples and overlays both the student input and instructor-provided sketch, and a total distance metric is calculating by summing the Euclidean distance between sampled points.

aperture. When the student clicks "Submit" after identifying the aperture area, FossilSketch checks to see if the location of the click is within the provided rectangular region. If it is, it is marked as correct. The location of the aperture is only used for identifying a foraminifer's morphotype.

The **summary screen** (step 6 for Foraminifera, and step 8 for Ostracoda) appears as the last step for each identification exercise, asking the student to draw from their observations and make the final selection of the foraminiferal morphotype, or Ostracoda genus. Each foraminiferal morphotype or Ostracoda genus has a list of characteristic features, and based on student answers, each feature

correctly marked during the identification steps would have a blue check-mark. Choices of foraminiferal morphotypes, and Ostracoda genera are ranked by the highest number of matching properties with student answers. If student answers are correct, the choice is easy since it has the most check-marks and is the first item listed. Additionally, a picture of each choice is included, letting students double-check to see if their best-ranked choice is the most accurate. This system allows students to develop self-assessment skills to see if their choices match up with any given morphotype or genus. At any time students are able to revisit any of the previous steps, so this final choice would be a good motivation to do so if they notice their prior choices did not yield a definitive conclusion. It also allows students to see different properties that might be common between some morphotypes or genera, but each foraminifer and ostracod will have only one correct final answer.

### 3.2.5   Assessment exercise

Once the students gain mastery of microfossil identification through practicing mini-games and microfossil identification, they proceed to the final type of exercise and assessment where they can apply their knowledge to reconstruct environments from an assemblage of different microfossils. In this exercise, the students view microfossil assemblages with approximately 20 foraminifer or ostracod individuals and identify the foraminiferal morphotypes or Ostracoda genera present. These assemblages imitate an actual microfossil "slide", as seen under a microscope that contains an assemblage of Foraminifera or Ostracoda. Students are asked to identify how many of each foraminiferal morphotype or ostracod genus are present in the slide. Before students start working on the exercise, they can view a screen with a summary of the information on foraminiferal morphotypes or ostracod genera and how they can be used to interpret environmental properties, such as the oxygenation or salinity of the water. This exercise includes 3 rounds and a summary. The student then needs to identify the different genera or morphotypes and select from the menu on the right side of the screen the number of each morphotype. It is intended that students will draw on their knowledge from the previous exercises to quickly identify the morphotypes or genera they see in these assemblages. For the ostracod assemblages, the menu to select from includes both the genera that are and genera that are not present in the assemblage. For the foraminiferal morphotypes, the assemblage includes two morphotypes to select from and "Other" category. To answer correctly, the student must provide a correct number for all categories, that is the two morphotypes or "other" for Foraminifera or genera for Ostracoda in an assemblage.

Both assemblage exercises conclude with a summary page where the student is asked to make an overall conclusion about the environment based on the assemblages. For instance, the Foraminifera morphotype assemblage exercise uses assemblages to determine for bottom water oxygenation. It has been shown that in environments where cylindrical- and flat-tapered morphotypes are found in abundance, the environments usually have low oxygenation [30]. The students are asked to rank each assemblage by relative oxygenation level. They should be able to do so when they consider the relative abundance of cylindrical-tapered and flat-tapered morphotypes they found in each of the three assemblages. Similarly for Ostracoda genera, students count the number of individuals of each genera, and determine the bottom water salinity indicated by each of the assemblages. These exercises assess microfossil identification learned and honed across all exercises of the FossilSketch system, and shows how microfossil research is applied.

## 4   EVALUATION

FossilSketch was deployed as part of a laboratory exercise in a class titled "Life on a Dynamic Planet" for Fall 2021 at the investigator's university. [co-author name redacted for review] is the

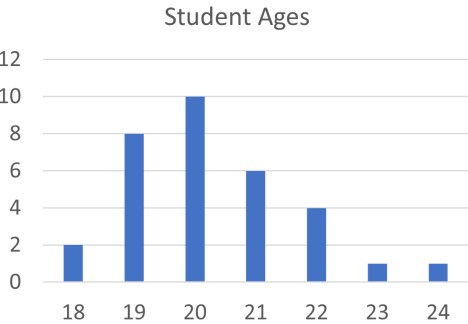

Figure 8: Distribution of student ages among those who consented to have their age information included in the study.

course's instructor for this class, and she introduced the students to the FossilSketch system. Students were instructed to watch the educational videos before coming to class. During the lab, they went through Foraminifera mini-games, morphotype identification and assessment exercise modules. Ostracoda modules were offered as an extra credit.

### 4.1   Design Study

Over the course of two weeks, a total of 32 students were asked to complete their assignment. All students were instructed to use the FossilSketch system as part of their assignment but consent to provide us data (surveys, focus group and sketch data) was fully optional. A total of 22 students consented to provide us data on their usage of FossilSketch for analysis.

#### 4.1.1   Study Population and Informed Consent

This study conformed to the university's Institutional Review Board protocol, IRB2019-1218M (expiration date 02/09/2023) ensuring the data is published only on users who gave us informed consent. Consents were distributed on the paper during the introductory portion of in the laboratory session. Of the 22 students who gave consent to have their demographic information published, 13 provided data on their race/ethnicity: 8 were White, 3 were Hispanic, 1 was Black, and 1 was Asian. Student ages ranged from 18 to 24, with specific age distribution shown in Figure 8

#### 4.1.2   Data Collection Protocol

The first module in FossilSketch has students complete a pre-study questionnaire that requests basic demographic information, and information on prior experience with micropaleontology and the topics covered in the FossilSketch interface, interest and self assessment in micropaleontology skills, and interest in future careers in micropaleontology. Similarly, the final module in FossilSketch is a post-study questionnaire that repeated questions regarding self-assessment of skill, interest in future careers involving micropaleontology, and feedback on use of FossilSketch. Most of the questions used a five-point Likert scale questions, and to provide us feedback students could elaborate in free-response forms. At the conclusion of the study, students were asked for feedback on their experience with the FossilSketch UI as part of informal interviews using focus groups with a subset of participants who were selected based on their agreement to take part on the focus group interviews.

FossilSketch tracks student performance by recording a student's "star rating" for each submitted exercise in an off-site grade-book SQL database. As a reminder, the final score of all exercises in FossilSketch is a rating ranging from one to three stars, with one being the most error-prone performance and three being error-free. Students are encouraged to repeat exercises if they did not receive

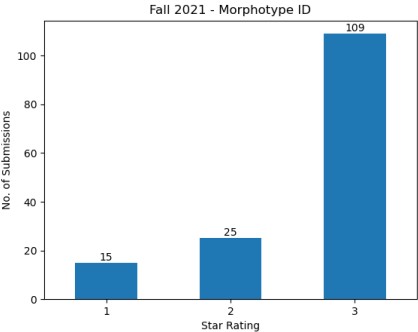

Figure 9: Visualization of the star ratings of submissions across all students.

| | | Rating (Stars) | | |
|---|---|---|---|---|
| | **Activity** | **1** | **2** | **3** |
| **Foraminifera** | Identification | 15 | 25 | 109 |
| | Morpho Match | 0 | 6 | 5 |
| | Chamber Match | 0 | 7 | 8 |
| | Assemblage | 0 | 5 | 6 |
| **Ostracoda** | Identification | 0 | 4 | 59 |
| | Orientation | 0 | 0 | 6 |
| | Outline | 0 | 6 | 3 |
| | Assemblage | 0 | 13 | 0 |
| **Totals** | | **15** | **66** | **196** |

Table 1: Number of student submissions for each FossilSketch activity.

three stars, and the website records every completed attempt in the grade-book database. This information lets us gauge overall performance in student activity on a per-exercise basis, and combining these responses with the more qualitative responses from students during focus group interviews and post-study questionnaires lets us analyze student interest.

## 4.2 Results

Study data can be summarized as "Quantitative" and "Qualitative", with the former being the recorded performance metrics found in the grade-book SQL database and the latter summarizing student sentiment about the FossilSketch user experience.

### 4.2.1 Quantitative

Modules that were tracked included all exercises and assessments, but activity on viewing videos was not tracked. However, the FossilSketch layout first displays the video modules, and instructors verbally encouraged students to complete the site's modules in order. The activities that were tracked in the grade-book SQL database are: foraminifera chamber matching game, foraminiferal morphotype matching game, morphotype identification exercise, assessment - paleoreconstruction using morphotypes, ostracod orientation game, ostracod outline matching game, ostracod genera identification exercise, and an assessment - ostracod assemblage exercise. Details on the exercises can be found in Sections 3.3 and 3.4. As a reminder, in this activity students were only required to complete the foraminifera exercises, with ostracod exercises existing as optional extra credit activities.

Table 1 summarizes the student submission data during our study. Figure 9 shows the submission score for the star ratings for the Morphotype Identification exercise. As expected, ostracod exercises

received fewer submissions due to the extra credit nature of the exercises. However, it should also be pointed that both ID exercises received a much higher volume of submissions due to the module requiring at least 3 submitted foraminiferal morphotypes (out of a possible 17) and 3 submitted ostracods (out of a possible 10). If students further submitted one of the three but decided to retry for a better score, it would be counted as another submission. Out of a total of 32 students, this means students submitted an average of 3 submissions of ID exercises per student for foraminifer morphotypes, and 7 submissions per student for ostracods of the students who chose to complete the extra credit (a total of 9 students completed the extra credit modules).

### 4.2.2 Qualitative

**Surveys and lab assignments feedback**. The following feedback was requested from students: 1. On a scale of 1 to 5, with 1 being completely disagree, and 5 being completely agree, how would you respond to the statement "I enjoyed the micropaleontology activities in this class." Please provide at least one example to explain your answer.

The most common rating the students gave was 3 (n=11). Most of students pointed to some software bugs and this is likely why few people rated it 4 and 5. Students' open-ended comments indicated that they: *"enjoyed the identification aspect of the activities that allowed me to investigate and figure out where a sample fossil was found." "it was very buggy and that made it frustrating but the overall system was a good way to learn."*

2. Did you work on the micropaleontology activities outside of class (other than class time)? If so, please explain what you did.

Approximately 50% of the students completed activities outside the class. Students' answers indicated that many of them used FossilSketch to finish lab assignment at home: *"I did not finish in class so I completed the assignment at home." "yes, I watched videos and checked my lab answers." "Yes, I just finished the lab on my own time."*

3. How did you feel, typically, while you were working on micropaleontology activities in this class?

Eleven students provided the answer, five indicated that activities were enjoyable, and six people felt it was confusing since they did not have prior knowledge.

4. Do you think the micropaleontology activities in this class are and will be useful to you? How so?

More than half of the students (n=10) who provided answers said that micropaleontology activities in this class are useful for future work, and career. The following quotes were associated with these answers: *"yes, I am a geology major so I will likely use this later in school and in my career." "Yes, because I would like to go into paleontology as a career (although not micropaleontology), so it would be good to have prior knowledge in these areas."*

5. When did you feel uncertain or unsure about something while working on micropaleontology activities in this class? How did you deal with this uncertainty?

The most common answer (n=7) was that a student went back to FossilSketch to look for answers.

6. What was helpful in FossilSketch activities?

Students almost unanimously (n=12) said that videos and mini games were very helpful. The following quotes were associated with this question: *"The videos and games."; "yes, I watched videos and checked my lab answers." "It was difficult to remember everything, so I went back in the videos and games." "Practice with identification" "Videos helped a lot with the lab questions." "YT videos + mini games" "The games were quite difficult, I rewatched videos and replayed the games until I was confident." "The videos were the most informative" "The videos and minigames were very helpful in explaining the different morphotypes".*

| Resource Type | Count |
|---|---|
| Rewatched FossilSketch videos | 16 |
| Retried FossilSketch games | 14 |
| Retried Morphotype ID games | 12 |
| Collaborated with others | 9 |
| Used in-person handouts | 5 |
| Other | 2 |

Table 2: A count on the different resources that students used to complete their lab assignment.

Additionally, when completing their lab assignment, students were asked what resources did they use to answer questions about microfossils. The following table shows that students used FossilSketch activities for completing their lab assignment, with videos, mini games and morphotype ID being the most common.

**Focus group feedback** In the focus groups discussion, students provided the following feedback:

1. How was your experience using FossilSketch?

*"Good. The website was easy to navigate. There were no crushes and bugs. Learning material was easy to access. I like how we could learn with the videos, but I do wish that videos also had slides to go back to individually rather watching the entire video."; "It was good, the videos were good, the games were cool." ; "I liked the games and that we could re-try them until we've learned."*

2. Anything you disliked? *"I wish we had feedback to know what we did wrong instead of just saying "it's wrong".": "Sometimes it was buggy, zooming in and out didn't work."*

3. If you were to add new features to FossilSketch, what would it be?

*"The games need hints for correct answers."; "Review sheet for the videos would help."; "For the stars, add percent, or partial stars, like 3.5."*

4. If you were to take another class would you want to use FossilSketch, or be in traditional class without software?

*"Prefer to use software, creative applications make learning easier."; "FossilSketch could be supplementary to traditional classes. The best would be to combine."*

5. What was your favorite activity in FossilSketch?

*"Morphotypes identification game."; "I like the extra credit (Ostracoda) activities, they were easier than the main ones."; "I liked the videos, they were the most informative."*

6. For sections with mini-games, morphotype identification and the paleoreconstruction assessment, the first time you worked with it, did you know what to do? Was it intuitive?

Majority of the students reported it was intuitive and they did not have any problems navigating between different steps of each section.

## 4.3 Discussion

We observed a measurable amount of student interest across FossilSketch submissions overall via a combination of analyzing exercise submissions and qualitative results, although we will also note there were varying degrees of interest when observing individual exercises and games. Morphotype and Genus ID exercises for both types of microfossils comprised the highest number of submissions by a wide margin, with lower observable numbers of submissions in template matching and environmental reconstruction games for the required portion of the lab assignment. There were a total of 15 submissions for the chamber matching game, and 11 submissions for the environmental reconstruction out of a total of 32 participants who used the system in the class (see Table 1). For morphotype ID, the 149 total submissions is partially explained by the requirement of completing 3 submissions as part of the lab exercise, but that alone does not account for all submissions since students submitted an average of 4.66 submissions. One possible inference is that students felt encouraged to complete the ID exercises in particular because the design of these activities was more appealing, an observation we found important due these exercises being the most complex in Fossilsketch. As section 3.4 specifies, ID exercises consisted of several interactions including sketching, pointing, and completing multiple choice questions over 6-8 separate steps, which offer cumulative observations about the morphotype or genus in question. By contrast, the matching games consist of one main interaction and do not involve the student drawing a conclusion. We believe the engaging design and applied problem solving implemented into the ID exercises can be accounted for the increase in the number of total submissions and average submissions per student, well above the required three per student.

Qualitative feedback was overall positive with various students indicating intuitive user experience. Some students specifically mention the identification exercises as the activity they most enjoyed. Students rated videos, games and ID exercises as very useful when completing the lab assignment. Some students mentioned they found the games initially difficult and others consider the subject of micropaleontology to be difficult in general, but were able to improve their understanding of the subject by referring to the informational materials in FossilSketch and rewatching videos and repeating exercises in the system. Students were also able to complete the lab assignment remotely at home, which would not have been possible in a traditional lab environment without FossilSketch. Table 2 lists the student answers for resources used to complete the lab assignment, with 42 of 58 answers (72%) using either FossilSketch videos, games, or ID exercises for assistance in their assignment.

The primary difficulty in interactions was the lack of scaling in the FossilSketch interface, which made certain low-resolution or zoomed-in displays leave out UI elements that made it difficult to complete the exercises. Some students would change the zoom level of their screen, which would result in the "bugs" that some students mentioned in their qualitative feedback. Some students also expressed disinterest in the system largely due to micropaleontology not being relevant to their major of study.

Overall, we observed that the proposed system was successful in providing an engaging and informative tool for learning that students were interested in using on their own to complete the class's laboratory assignment. Generally positive feedback from students and a large number of submissions for the identification exercises suggest a positive overall learning experience and succeeds in our goal of an intuitive educational tool that can be used in tandem with in-class learning.

## 5 FUTURE WORK

The modular design of FossilSketch provides flexibility in creating course-specific landing pages, so we will continue to iterate on the existing exercises for additional polish and bug-fixing reported in the study. Additionally, we intend to implement an instructor interface that would provide instructors with a login that would display their student submissions and performance. In addition, this interface will provide instructors with a system to create their own landing pages from within the website, allowing them to alter the order, add or remove exercises. We also intend for this interface to allow instructors to add more Foraminifera and Ostracoda morphotypes and genera for the identification exercises. We expect that these additions will allow this system to be deployed in various classrooms with a large number of instructors without the need for web developers to implement changes for each instructors' needs.

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
