# OpenReview forum: "FossilSketch: A novel interactive web interface for teaching university-level micropaleontology"
_graphicsinterface.org/Graphics_Interface/2022/Conference — Submitted to GI 2022_

### Official Review · Reviewer_YU1w · 2022-01-12
**This investigation of a specific e-learning system needs major revisions before it will be ready for publication**

**Rating:** 4
**Confidence:** 5

**Review:**

This paper presents a new e-learning system and the algorithms it uses to support a specific type of learning for the geosciences (i.e., micropaleontology). I have major concerns about the reporting of the methods, results, and consequently the amount of support for various claims. For these reasons, I cannot recommend acceptance at this time.

There is relevant literature from other areas of e-learning that should be considered. This includes visual-perceptual training research and research on intelligent tutoring systems. Please see below.


Assumptions and claims that need more support:
- That encouraging repeat submissions is good (see work on gaming the system)
- that the system is an effective learning tool (no evidence of learning was provided; learning is defined as a change in knowledge, skills, or perspectives/attitudes)
- "This system allows students to develop self-assessment skills to see if their choices match up with any given morphotype or genus" As far as I can tell, self-assessment skills were not measured.


Methods:
- the study methods seem to described across the paper in different locations (e.g., 3.1, partially embedded in the results)
- In 3.1, please describe the expertise of the co-author who has expertise. This is needed for readers to assess the quality of those aspects of the work
- In 3.2.4, a "scaffolded learning experience" is claimed. In education, scaffolding requires the gradual removal of support until a student can do a task independently. How was this achieved?
- In 3.2.4, the interactions are described as achieving a goal (e.g., help student retain) rather than as their having the intent of doing that. These claims need to be reframed since these are the goals of the system and have not been shown to be true (especially at the system design stage)
- When describing the system contribution (algorithms), the process for empirically determining values must be detailed.
- How was mastery defined for the purposes of this system and study?
- How were participants remunerated (if at all)?
- Report the rating scale anchors for questionnaire items. Also, it's unclear from the methods section whether Likert or Likert type items were used. The paper describes Likert scale questions. Technically, a Likert scale cannot be used with a question because it indicates agreement. Likert-type rating scales can be used with questions. Based on later reporting, I suspect this is a wording issue and that questions needs to be changed to items in the methods description.
- The human-subjects study analysis procedures are not reported in the methods. Some are implied or briefly described in the results.
- Why was video viewing not tracked?
- Data aren't qualitative or quantitative. I realize quotes were used for this, but the later results break from this framing by mixing quotes with numerical questionnaire ratings in the qualitative section. Organizing the results around the research question helps to avoid this problem.



Results
- why were raw counts used for reporting when the opportunity for performing certain activities varied?
- always report a measure of variability when reporting a central tendency (e.g., SD when providing M)
- the last paragraph of 4.2.1, references 32 students. I had understood that 22 consented. This is either unethical from an REB perspective or a mistake that is misleading for readers (for the purposes of my review, I've assumed it's a reporting mistake)
- show the questionnaire results with a divergent stacked bar chart so that the distribution of responses is obvious to readers
- I didn't see anything about drawing in the user study results despite a large portion of the contribution being focused or framed around this
- the results of the focus group need more interpretation and synthesis
- the interpretation of some of the behavioural findings should account for novelty effects and similar behaviour patterns that are common in e-learning



Literature:
- Koedinger, K. R., Corbett, A. T., & Perfetti, C. (2012). The knowledge-learning-instruction framework: Bridging the science-practice chasm to enhance robust student learning. Cognitive Science, 36(5), 757–798. https://doi.org/10.1111/j.1551-6709.2012.01245.x
- Hoven, S., Seniuk, A., Martel, M., Khan, H., Rourke, L., & Demmans Epp, C. (2018). Adaptive Visual-Diagnostic Training: User Mental Model Development. 44–45.
- Rourke, L., Oberholtzer, S., Chatterley, T., & Brassard, A. (2015). Learning to Detect, Categorize, and Identify Skin Lesions: A Meta-analysis. JAMA Dermatology, 151(3), 293. https://doi.org/10.1001/jamadermatol.2014.3300
- Xu, B., Rourke, L., Robinson, J. K., & Tanaka, J. W. (2016). Training Melanoma Detection in Photographs Using the Perceptual Expertise Training Approach: Training melanoma detection in photographs. Applied Cognitive Psychology, 30(5), 750–756. https://doi.org/10.1002/acp.3250
- Baker, R. S., Corbett, A. T., Roll, I., & Koedinger, K. R. (2008). Developing a Generalizable Detector of When Students Game the System. User Modeling and User-Adapted Interaction (UMUAI), 18(3), 287–314. https://doi.org/10.1007/s11257-007-9045-6
- Beck, J. E., & Gong, Y. (2013). Wheel-Spinning: Students Who Fail to Master a Skill. In H. C. Lane, K. Yacef, J. Mostow, & P. Pavlik (Eds.), Artificial Intelligence in Education (pp. 431–440). Springer. https://doi.org/10.1007/978-3-642-39112-5_44
- Demmans Epp, C., & Phirangee, K. (2019). Exploring mobile tool integration: Design activities carefully or students may not learn. Contemporary Educational Psychology, 59(2019), 101791. https://doi.org/10.1016/j.cedpsych.2019.101791
- Brooks, C., Demmans Epp, C., Logan, G., & Greer, J. (2011). The who, what, when, and why of lecture capture. Learning Analytics and Knowledge, 86–92. https://doi.org/10.1145/2090116.2090128
- Barokas, J., Ketterl, M., Brooks, C., & Greer, J. (2010). Lecture capture: Student perceptions, expectations, and behaviors. In J. Sanchez & K. Zhang (Eds.), World Conference on E-Learning in Corporate, Government, Healthcare, and Higher Education (pp. 424–431).

---

### Official Review · Reviewer_WbaG · 2022-01-12
**Although the paper presents an interesting active-learning teaching tool, it does not describe novel graphics or HCI techniques or evaluation.**

**Rating:** 5
**Confidence:** 4

**Review:**


This paper describes a web-based system that helps students develop skills in micropaleontology. The system consists of a web-based application that combined instructional videos, surveys, and sketch activities.

Strengths:

The system helps students actively engage with the material and develop micropaleontology skills. This system shows potential as an effective active learning complement to the authors' lecture and class materials.

Weaknesses:

GI may not be the right venue for this work. Although this paper is interesting from a pedagogical perspective, it does not present novel HCI and graphics material. The analysis is a pilot study to show proof of concept, not an evaluation of the effectiveness of the system for teaching.

Fixable weaknesses:

The paper could offer more details to help a reader replicate their work for their courses. Specifically,

- What is the web application written in (Javascript?, React? ,Qualtrix?)
- How/where was the application hosted?
- How is content added to the application?
- You mention gradebook SQL. Do you have a reference to an API for that?

The sketch feature is the most interesting graphical aspect of the platform but the paper lacks details to indicate how well this feature works.

- what is the resolution of the images?
- what is the effect of the thresholds that were chosen for the distance metrics? Specifically, how "wrong" does a drawing need to be before it is categorized as incorrect? Do you encounter situations where the system identifies an answer is incorrect but a human instructor would mark it as correct?

Algorithm 1 is unclear. My assumptions were

- path is a list of (xi, yi) pixel coordinates
- `i` is an index from 1 to numPoints-1
- new points are appended to `out`, not set equal

---

### Official Review · Reviewer_mrfi · 2022-01-17
**A Sufficiently Useful Case Study?**

**Rating:** 6
**Confidence:** 5

**Review:**

This paper describes the implementation and deployment experiences of FossilSketch, a sketching system that allows students to receive interactive feedback during exercises. In flavor, it is reminiscent of Joe LaViola's work on MathPad and other similar systems that have been deployed in classroom-based environments.

This paper is a very well written case study, and, for me, with some past experience in sketch recognition, it was sufficiently clear. Figures 5 and 6 were particularly helpful in understanding the workings of the system.

The challenge with this paper is, obviously, in the originality and significance of the work. The work is original per se, in that it is deployed in a paleontology lab, and I am not aware of any systems that are deployed in similar environments.  However, in contrast to work like that from Tom Stahovitch's group on electric circuit diagrams or from Joe LaViola on math exercises, what is unclear to me is how much variability can be created automatically in the exercises produced. For example, in Stahovitch's group's past work on electrical diagrams (De Silva, R., Bischel, D. T., Lee, W., Peterson, E. J., Calfee, R. C., & Stahovich, T. F. (2007, August). Kirchhoff's pen: a pen-based circuit analysis tutor. In Proceedings of the 4th Eurographics workshop on Sketch-based interfaces and modeling, pp. 75-82) the system infers characteristics of the electrical circuit, recognizes that math, and analyzes the correspondence to identify errors. Similarly, Mathpad2 (LaViola Jr, J. J. (2007). An initial evaluation of MathPad2: A tool for creating dynamic mathematical illustrations. Computers & Graphics, 31(4), 540-553.) allows smilar interactive sketches that require little intervention for students to draw, create, and explore independently. The current system feels like a lot of customization of problems must be done by the instructor a priori to create and execute exercises.

On the other hand, there is significant work on building tools to support learning through the completion of more engaging exercises or tasks. The entire field of edugames is essentially this: a series of exercises designed and engineered to foster student learning through something that is more fun and engaging that some abstract set of pen and paper tasks. This isn't really similar to that work -- there is limited gamification here -- but it does define this middle-ground space.

What would be particularly useful for making this middle ground better explored is a more critical review of the design. In the current paper, we are treated to relatively shallow information (probably from a questionnaire) based on experiences students had completing the lab exercises, plus, essentially, the answers to six questions during the focus group sessions. What would make this case study more useful would have been a series of steps in review to critically assess the work, including:
- Thinkalouds, perhaps in small groups. I would love to understand how students found the exercises and the supports for the exercises while they were doing them. It isn't clear if there is interactivity and feedback that is better than what one would get with an exercise book. Was this different? Did students receive real time feedback? Or was it basically saving paper by making the exercise digital and some simplification of grading? I believe feedback was provided to students, perhaps in real time, based upon the statement in the paper "Students are encouraged to repeat exercises if they did not receive three stars ..."
- Some reflection by the professor on experiences with FossilSketch. I realize that this presents a challenge when the professor is also an author, but I would point to work like Neudstadter's and Senger's on autobiographical design (Neustaedter, Carman, and Phoebe Sengers. "Autobiographical design in HCI research: designing and learning through use-it-yourself." In Proceedings of the Designing Interactive Systems Conference, pp. 514-523. 2012.). I believe there are ways to critically reflect on a system, even when roles are combined as author and user of the system. All qualitative work is, in some sense, subjective. Ethnography, itself, is a participatory exercise. How did the system help? What more is needed?
- A deeper focus group discussion, and a broader demographic of focus groups. The focus group results were disappointingly short. Was it just answers to questions, or did they involve broader discussions. Why not semi-structured in interview format and feel. What themes were brought up. Also, the limitation to students is problematic. Beyond the above point on autobiographical design, I am curous as to thether there wereere TAs in the course? Do other instructors teach the course? ARe there other types of participants that could also participate in a focus group?

Finally, I note that this system was deployed in 2021, in the middle of a global pandemic. Was this really deployed in an in-person lab setting? And, given the pandemic, are there ways that it potentially helped with the lab component? Or was the lab distributed? I see little in the lab that requires in-person instruction, so maybe there was this distributed component to the system, and, if there was, this, too should be reflected in results. Even if the lab was done in person, I assume there were some capacity restrictions, issues of masking, distancing, and absences due to illness. Did this system help with addressing that?

At the end of the day, this paper represents a judgement call. The paper could, to me, have been much more. Its current addition to the literature is really quite minor. On the other hand, the litmus test that I typically use for Graphics Interface is to ask myself whether the paper I am reading would be something useful for a Masters student to read, such that they could learn something and use it to begin research contributions. Given this standard for comparison, I believe that this paper is just over the bar.

---

### Decision · Program_Chairs · 2022-01-18

Reject